# Diagnosis and Management of Pelvic Venous Disorders in Females

**DOI:** 10.3390/diagnostics12102337

**Published:** 2022-09-27

**Authors:** Clément Marcelin, Yann Le Bras, Isabelle Molina Andreo, Eva Jambon, Nicolas Grenier

**Affiliations:** Department of Radiology, Centre Hospitalier Universitaire (CHU) Pellegrin de Bordeaux, 33000 Bordeaux, France

**Keywords:** pelvic venous, pelvic pain, congestion syndrome, embolization

## Abstract

Pelvic venous pathologies in females are responsible for chronic symptoms grouped under the term pelvic congestion syndrome, which includes chronic pelvic pain, perineal heaviness, urgency, and postcoital pain, along with vulvar, perineal, and lower limb varicose veins. These conditions are also associated with ovarian and pelvic venous reflux and venous obstruction. This review aimed to explore the clinical and imaging modalities for diagnosing pelvic congestion syndrome, pelvic venous pathologies, their therapeutic management, and their outcomes.

## 1. Introduction

Pelvic venous insufficiency is a common pathology that affects approximately 8% of females of child-bearing age and is the cause of 20% of gynecological consultations for chronic pelvic pain [1]. It results in pelvic congestion syndrome (PCS), which has symptoms such as chronic pelvic pain, dyspareunia, dysmenorrhea, lower limb venous insufficiency, and vulvar and perineal varices [2,3,4]. These symptoms have recently been grouped under the term “pelvic venous disorders” (PVDs) [5]. This pathology, which has previously been underdiagnosed, is increasingly being recognized. The Symptoms-Varices-Pathophysiology classification proposed by the American Vein and Lymphatic Society International Working Group on Pelvic Venous Disorders [5] aids clinical and paraclinical diagnosis of PVDs and allows administration of the best treatment.

Gynecological clinics are responsible for PVD diagnosis and must be able to differentiate chronic pelvic pain associated with PCS from other gynecological, urological, neurological, and gastroenterological etiologies. Various invasive and noninvasive diagnostic modalities are available for varicose veins and obstructions. Three important reservoirs wherein varicose veins develop must be investigated using imaging modalities (Figure 1): the kidney hilum (zone 1); the venous plexus and ovarian and iliac veins (zone 2); and the extrapelvic perineal and vulvar region (zone 3) (Figure 1). Noninvasive modalities, including ultrasonography (US), magnetic resonance imaging (MRI), and computed tomography (CT), can also exclude other pelvic pathologies; however, their sensitivity for diagnosing pelvic insufficiency remains low, at 59%, 20%, and 13%, respectively [6]. Therefore, venography remains the modality of choice. However, interpretations vary and catheterizing pathological afferents is difficult, particularly at the pelvic level [7,8]. Venography is also invasive and generally precedes therapeutic interventions. Several options have proven to be effective for pain resolution, including medical treatment, surgery, and percutaneous embolization, which has become the standard treatment for pelvic venous insufficiency and results in symptomatic improvements in 80–94% of patients [9,10,11].

### Physiology of PVD

PVDs are mainly caused by pregnancy, and in rare cases, ovarian and internal iliac venous insufficiency occur due to valvular incompetence or compression of the left renal or left common iliac vein. Hormonal impregnation of estrogens and progesterone causes significant dilation, loss of elasticity of the venous wall, and separation of the valve walls, which are responsible for venous reflux and failure to return to normal in the months following childbirth. The ovarian, pelvic, and extrapelvic systems are highly interconnected. The ovarian system is connected to the uterine system, directly and bilaterally, through the pampiniform transverse drainage veins. The uterine system interacts with pudendal, obturator, and buttock veins, which play roles in incompetence. In a state of stress or distention, pelvic leak may occur (Figure 2). This can involve what we term here an “inguinal vanishing point” (round ligament vein), as well as obturator, pudendal, and gluteal veins. Vulvar varices are the most common manifestation, but other perineal and buttock regions may be involved. Symptom type and location depend on whether pressure is transmitted to the “distal venous reservoir” only (“uncompensated obstruction”), and on whether decompression (“compensated obstruction”) occurs due to reflux. Uncompensated obstruction in ovarian or internal iliac vein tributaries may lead to chronic pelvic pain. Reflux may be “decompressed” through “pelvic escape points”, which affects the lower extremities and results in vulvar varicosities without pelvic pain. Similarly, uncompensated obstruction of the left renal vein or common iliac vein leads to pressure and symptoms in the left kidney or lower extremities. Compensated obstruction of the veins caused by reflux (through the ovarian or ipsilateral internal iliac artery) results in pelvic venous hypertension and varicose veins and/or lower extremity and vulvar varicosities.

## 2. Clinical Diagnosis

There are four clinical presentations of pelvic venous pathology.

### 2.1. Left Renal Vein Symptoms of Venous Origin

These are related to renal vein compression (nutcracker syndrome) and can manifest as left lumbar, abdominal, and pelvic pain, with microscopic or macroscopic hematuria.

Renal vein compression (nutcracker syndrome) symptoms, i.e., hematuria proteinuria and lumbar and pelvic pain [1], are associated with increased pressure on intrarenal venous structures. This “hyper pressure” is responsible for the development of varicose veins in the pyelic and periureteral regions, where cavity rupture can cause hematuria [2]. Within the nephron, venous hypertension is responsible for orthostatic proteinuria. In a more advanced stage, left renal venous drainage occurs via the lumbar venous plexus and left ovarian vein. Dilation and reflux of the left ovarian vein are responsible for pelvic congestion syndrome.

### 2.2. Chronic Pelvic Pain of Venous Origin

This is defined as chronic pelvic pain associated with urinary, digestive, or sexual problems. The American College of Gynecology and Obstetrics [12] has proposed the following criteria for chronic pelvic pain: noncyclic pain present for at least six months, localized in the pelvis, anterior abdominal wall, and at or below the umbilicus, or disabling lumbosacral or buttock pain requiring medical attention. The prevalence is estimated to be between 14.7% and 24% in females of child-bearing age. Apart from chronic pain, there is also an alteration in the quality of life. Pain is associated with one or more abnormalities, including endometriosis, adenomyosis, pelvic inflammation, interstitial cystitis, irritable colon, fibromyalgia, chronic low back pain, neurological compression, and pelvic venous disease [13,14].

To differentiate chronic pelvic pain related to PCS from other etiologies, an initial clinical assessment form is often helpful. A recent study [15] suggested 10 clinical parameters for the evaluation of PCS using a Visual Analogue Scale (VAS) for pelvic pain, rated from 0 to 10, which are dyspareunia, postcoital pain, periods, pain in the lower limbs, difficulty walking, aesthetic problems, impacts on work, psychology, and daily activities. Typically, PCS affects females aged 20–45 who are often multiparous. Pain is significant, disabling (dull and aching in character), noncyclical, and aggravated by standing and walking, and can be more intense at the end of the day and during menstruation. There is also peri- and postcoital pain. In most cases, there is an alteration in the quality of life (due to lethargy and depression). Other manifestations are similar to other pathologies, including gynecological (dysmenorrhea), urological (dysuria and urgency), and digestive (nausea, bloating, and abdominal and rectal pain) problems. The combination of postcoital pain and ovarian tenderness on clinical examination has a sensitivity of 94% and a specificity of 77% for the diagnosis of PCS [16].

### 2.3. Perineal and Lower Limb Venous Disease of Pelvic Origin

Pelvic venous pathologies may cause perineal, vulvar, and posterior thigh varicose veins in addition to the classic great saphenous varicose veins and are associated with a broad spectrum of lower limb venous pathologies and the CEAP classification (clinical-etiological-anatomical-pathophysiological). Perineal and vulvar varices are responsible for local symptoms, such as pain, heaviness, pruritus, and superficial thrombosis. Varicose veins of the lower limbs may be of atypical pelvic origin.

### 2.4. Proximal Iliac Vein Obstruction (May–Thurner Syndrome)

Extrinsic compression of the common iliac vein by the right common artery and the spine, known as May–Thurner syndrome, contributes to a secondary PSC and leads to edema and lower limb venous claudication.

## 3. Imaging for PVDs

The purpose of imaging is to identify the affected venous regions and the presence of venous obstructions or reflux.

### 3.1. Renal Region

The examinations must answer two questions: “Is there a significant compression of the left renal vein at the level of the aorto-mesenteric clamp (anterior nutcracker) or associated with the retro-aortic renal vein (posterior nutcracker)?” and, “Is the venous compression compensated (perihilar varices or reflux at the level of the ovarian vein)?”. Noninvasive radiological examinations, including Doppler US, CT, and MRI, may identify venous compression, but phlebography and venous pressure analysis are essential for definitive diagnosis [3]. Doppler US is used for anatomic and physiologic assessment, reference ratios are higher in the upright position, and peak systolic velocity (PSV) in the compressed area is higher compared to that in the hilum [17]. Ratios ranging from 4.0:1 to 5.0:1 are considered significant for nutcracker syndrome [17,18], and cross-sectional CT and MRI images showing vessel diameter and the superior mesenteric angle (SMA) can be used to identify significant left renal vein compression. A normal aorto-SMA is 45–90°; an angle of ≤35° suggests nutcracker syndrome [5]. The ratio between the diameter of the LRV at the renal hilum and aortomesenteric (AM) can be determined in the axial plane on CT. On CT, for a hilar-to-AM diameter ≥4.9 mm, 66.7% sensitivity and 100% specificity have been reported [5,6]. However, left renal venous compression >50% is seen in 51–71% of asymptomatic patients on CT [17]. Venography with venous pressure measurements remains the standard examination for renal venous hypertension with a pressure gradient >3 mmHg between the renal vein and the inferior vena cava [18].

### 3.2. Gonadic Veins, Pelvic Venous Plexus, and Iliac Intern Veins

The purpose of imaging the ovarian veins is to identify venous dilation and reflux. Spontaneous or induced reflux in the left ovarian vein is found in all patients with congestion symptoms, compared to 25% of asymptomatic patients [19]. The positive predictive values for 5.6 mm and 7 mm diameters are 71.2, 81.8, and 83.3, respectively [18]. Velocity-encoded phase-contrast imaging and time-resolved MR angiography allow the detection of reflux and measurement of venous diameter, similar to venography [20]; these modalities are more sensitive and specific than CT [21]. Right ovarian vein reflux is rarely detected, and pretreatment exploration using venography is recommended when the venous diameter is >7 mm [22,23].

Asciutto et al. compared dynamic MRI and phlebography in the study of 23 patients with PCS. The sensitivity and specificity of MRI were 88% and 67% for ovarian veins, 100% and 38% for hypogastric veins, and 91% and 42% for periuterine varicosity, respectively [24]. Yang et al. compared dynamic MRI and phlebography for diagnosing reflux in the ovarian veins in 19 patients: the sensitivity, specificity, and diagnostic accuracy was 66.7%, 100%, 78.9% for MRI, and 75%, 100%, and 84.2% for phlebography, respectively [20]. Meneses et al. used phase-contrast velocity mapping to diagnose PCS in nine patients, and reported a sensitivity of 100% and specificity of 50% [25].

Transabdominal and transvaginal US is the first-line modality for the diagnosis of pelvic varices of the ovarian venous plexus. The following criteria are used for pelvic varices: dilated and tortuous periuterine and perivaginal veins greater or equal to 6 mm, slowed or reversed flow, induced or noninduced, and dilated venous plexuses communicating via transuterine veins, which are also dilated [26]. CT and MRI are useful for the identification of pelvic varices as tubular tortuous structures with diameters ≥5 mm, enhanced by contrast or hypersignal STIR sequences (short TI inversion recovery). Reflux in the internal iliac veins or their tributaries, spontaneously or during Valsalva maneuver, can be demonstrated on transvaginal and transperineal US venography (transcatheter venography) of the ovarian veins and internal iliac veins [27]. However, it is an invasive, irradiating, and time-consuming procedure. Noninvasive examinations are used for the diagnosis of pathological veins and/or reflux, while venography is only performed preoperatively.

Laparoscopy has a sensibility of 40% in the detection of pathologies associated with PCS [28]; however, pelvic MRI has a better diagnostic value [14].

### 3.3. Perineal Varices of Pelvic Origin

Echo-doppler is required to map the perineal venous area and determine the most suitable treatment [27]. When posterior vulvar and paravulvar varicose veins are observed, pudendal, obturator, and gluteal veins should be checked for leakage [29]. Franceschi et al. described six “communication points” between the pelvis and lower limbs [30]: the perineal point (Point P, involving reflux of the medial pudendal vein), obturator point (Point O, involving reflux of the obturator vein), superior gluteal point (GS Point, involving reflux of the superior gluteal vein), inferior gluteal point (GI Point, involving reflux of the inferior gluteal vein), inguinal point (Point I, involving reflux at the level of the round ligament vein), and clitoral point (Point C, involving reflux of the dorsal vein of the clitoris). Varicose veins appear abnormal on MRI, with tortuous dilation seen on STIR sequences [22].

### 3.4. Lower Limb Varices

Pelvic venous insufficiency may cause primary or recurrent varicose veins of the lower limb. Duplex US is performed to assess the sapheno-femoral junction, great saphenous vein (GSV), and deep veins for valvular incompetence. Exploration of the lower limb veins using Doppler US should be a part of the evaluation of pelvic venous pathologies, particularly for atypical varicose veins (posterior and lateral thigh varices, and varicose veins in nonsaphenous regions) [31]. Varicose veins of the lower limbs originating from the vanishing points mentioned above suggest a pelvic origin. Reflux from Point P can be transmitted to the perineal veins and through Giacominini’s veins via intersaphenous anastomosis [32]. Postero-lateral varicose veins of the thigh develop through the GS Point. Reflux at the GI Point may be responsible for varicose veins of the sciatic nerve and are highly suggestive of a pelvic origin. Reflux at the level of the round ligament vein (Point I) is responsible for nonspecific (inguinal, saphenous, or extrasaphenous) varicosity but can also affect labial and perineal varices. Reflux at Point O is responsible for nonspecific varicosity of the lower limbs. In patients with a clinical history of PVD, if US demonstrates reflux in the groin originating from the superficial epigastric, pubic, or pudendal veins, along with atypical lower limb varicosity, PVD should be suspected, and pelvic Doppler US and MRI should be performed. Clinical and radiological examinations are required, following the recommendations of international societies [5,31], to classify each type of PVD before deciding on the treatment.

## 4. PVD Treatment

### 4.1. Nutcracker Syndrome and May–Thurner Syndrome

Several therapies have been proposed for left renal vein compression, including conservative management, surgical treatment (transposition of the renal vein, renal autografts, and/or transposition of the ovarian vein at the level of the common iliac vein), and endovascular treatment (stenting), but the level of supporting evidence is low [33,34].

Conservative management is sufficient in many cases, especially in the pediatric population. Most pediatric patients exhibit spontaneous resolution in association with retroperitoneal and/or mesenteric adipose tissue growth and the accumulation of fibrous tissue at the SMA [7].

Renal vein transposition is the most common surgical intervention for anterior renal vein compression. Laparoscopic and robot-assisted laparoscopic techniques are now implemented as standard [8]. Nevertheless, reintervention rates are as high as 68%. Left kidney autotransplantation is also relatively common and is considered to be the best approach when renal ptosis is present along with venous compression [9]. There is some evidence of the effectiveness of stenting for treating congestion syndrome and its symptoms [35] (Figure 3). Benefits of this approach include rapid recovery and symptom resolution. However, stent migration occurred in 6.7% of cases in a study with a mean follow-up of 45 months [10]. Iliac venous compression (May–Thurner syndrome) may cause congestion syndrome due to reflux in the ipsilateral iliac vein; in such cases, iliac vein stenting is the treatment of choice. In a retrospective study of 18 patients, Daugherty et al. demonstrated improvement in the average clinical venous score, from 7 (range: 0–10) to 3.5 (range: 0–9), after iliac vein stenting [36].

### 4.2. Ovarian and Iliac Venous Insufficiency

Venography involving retrograde catheterization of the ovarian and internal iliac veins is a prerequisite for percutaneous treatment of pelvic varicose veins. Venography requires venous access (brachial, jugular, or femoral) to catheterize the left renal vein and left ovarian vein; this is performed using a 4–5 Fr HH1 catheter (jugular or brachial access) or Cobra II catheter (femoral access). Next, catheterization of the right ovarian vein is performed with an HH1 or MP catheter (jugular and brachial access) or Simmons 1–2 catheter (femoral access). Finally, catheterization of the internal iliac veins is performed using an MP or HH1 catheter (jugular and brachial access), with a Cobra II, UAC (Merit Medical, South Jordan, UT, USA) or RUC catheter (Cook Medical, Bloomington, IN, USA) used for femoral access. Phlebography is performed during Valsalva maneuvers.

Percutaneous embolization is the standard treatment for the various pelvic venous pathologies, including PCS and venous insufficiency of pelvic origin (Figure 4). Coils, vascular plugs, sclerosants, glue, and ethylene vinyl alcohol copolymer liquid embolic agent (Onyx^®^, Medtronic, Dublin, Ireland) are commonly used for embolization. The embolization process differs depending on the venography results. At the level of the ovarian veins, a dilated vein ≥6 mm in diameter is associated with valvular incompetence, indicating the need for embolization. A microcatheter is recommended to treat parametrial varicosities and ovarian veins in cases with refluxing ovarian veins. Glue and Onyx make it possible to treat all afferents of an ovarian vein before mechanical occlusion or occlusion by liquids. Venography of the right ovarian vein should be performed if it is >6 mm in diameter and shows reflux in noninvasive examinations.

For varicose veins arising from iliac afferent veins, the Valsalva maneuvers can be used to detect incompetence and leak points. Pathological tributaries must be treated to improve PVD symptoms. Pelvic varicosities supplied by ovarian and iliac afferent veins must be embolized (iliac varicose veins followed by the varicose veins supplied by the ovarian vein). To address leak points near the lower limbs, a microcatheter can be used for embolization from the distal to proximal position, at the ends of the pathological afferents. This treatment should ideally be carried out in conjunction with the Valsalva maneuver. The Society for Vascular Surgery and American Venous Forum recommends the use of coils and sclerosing agents (level of evidence = 2B) [37]. For ovarian veins, metallic devices (coils or plugs) and 2% polidocanol (Aetoxisclerol, Kreussler, Germany) foam are typically used. Pathological iliac veins and pathological tributaries are treated with coils and/or sclerosant. The main issue when using mechanical devices is immediate or delayed migration. Thus, coils of appropriate size must be used. The use of glue in iliac varicose veins and leak points is effective; however, precise control is required. More recently, Onyx, used alone or in combination with sclerosants, has been proven safe and effective [15,38,39].

The effectiveness of embolization is evaluated based on the improvement in symptoms and psychological states thereof. Laborda et al. [40] used coils in 202 patients who were followed-up for 5 years. They reported a 93.85% clinical success rate, with complete resolution of symptoms in 33.52% of patients and a reduction in the mean VAS pain score from 7.34 ± 0.7 to 0.78 ± 1.2. Nasser et al. [41] reported a decrease in the mean VAS pain score from 7.34 to 0.47 at 12 months, with complete resolution of pain in 53% of their patients. In a large series of 520 patients who underwent embolization of the four main ovarian and iliac axes using coils and plugs, De Gregorio et al. [42] reported a decrease in the mean VAS score from 7.63 ± 0.9 to 0.91 ± 1.5 over 5 years. More recently, Senechal et al. [39] followed-up with 327 patients for an average of 39 months after embolization using Onyx and reported an improvement in the mean VAS pain score from 7 ± 2.4 to 1.2 ± 1.9. The clinical success rate was 92%, and the rates of dyspareunia and postcoital pain decreased from 50% to 19% and 69% to 11%, respectively. Onyx in combination with sclerosants [15] has also been proven effective, with clinical improvement seen in approximately 95.9% of patients (total improvement: 30.1%).

### 4.3. Vulvar and Lower Limb Varices of Pelvic Origin

Vulvar, perineal, or lower limb varices of pelvic origin have a lower than 10% association with PCS [43]. The contribution of embolization of pelvic varices, compared to the sclerosis of perineal and lower limb varices under US and/or fluoroscopic guidance, is controversial. A small series [44] was unable to demonstrate any benefit of embolization of pelvic varices and vanishing points in improving symptoms of lower limb varicose veins. There was, however, an improvement in the symptoms related to vulvar varices. Other authors reserve embolization for recurrences and refractory varicose veins in females without congestion syndrome [45,46]. However, it has been reported that an absence of prior embolization of pelvic varices and leak points in cases of lower limb venous insufficiency of pelvic origin leads to a higher rate of recurrence [47]. Symptomatic vulvar and perineal varices of pelvic origin can be treated by embolization of the pelvic varices and leak points. Sclerotherapy can be performed via direct puncture of the varices (Figure 5). Perineal varices of pelvic origin are often sclerotized after embolization if they persist. Persistence is linked to superficial and deep anastomoses, which are identified by perineal varicography and treated by puncture of one or more varicose veins under ultrasound guidance. Sclerosis is performed with injection of 0.5% or 2% sclerosant foam.

## Figures and Tables

**Figure 1 diagnostics-12-02337-f001:**
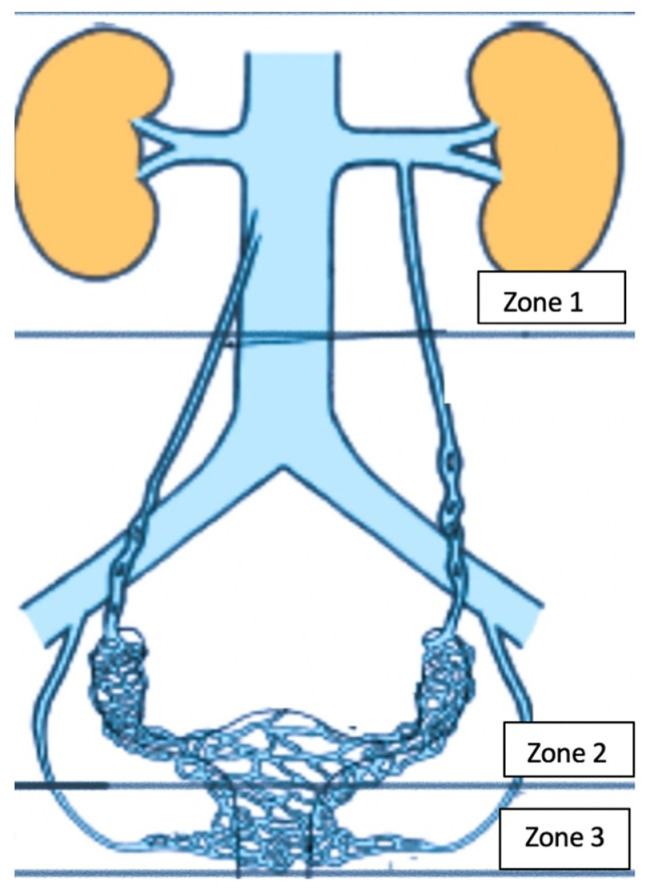
Clinical manifestations of pelvic venous disorders with involvement of three anatomical areas: Zone 1: left renal vein; Zone 2: ovarian and iliac veins plus the pelvic venous plexus; Zone 3: vulvar and perineal varices of pelvic origin.

**Figure 2 diagnostics-12-02337-f002:**
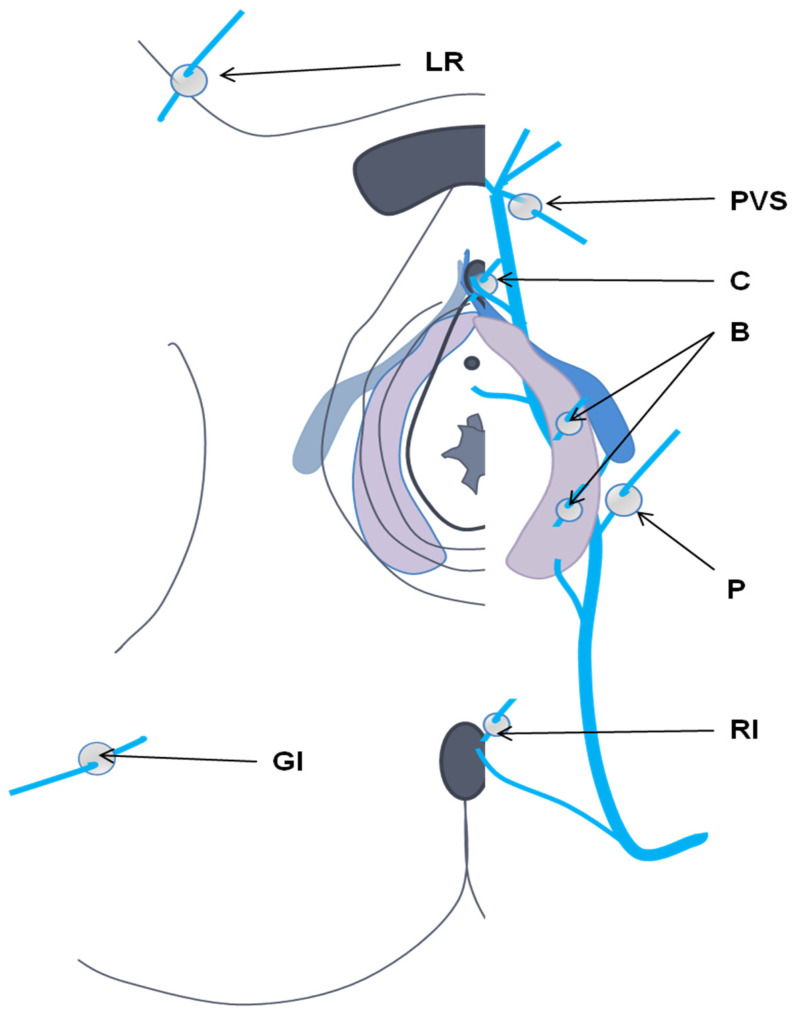
Cutaneous projection of pelvic leak points (perforation). LR, round ligament; PVS, venous plexus of Santorini; C, clitoral piercing; B, bulbar; P, perineal; GI, lower gluteal [12].

**Figure 3 diagnostics-12-02337-f003:**
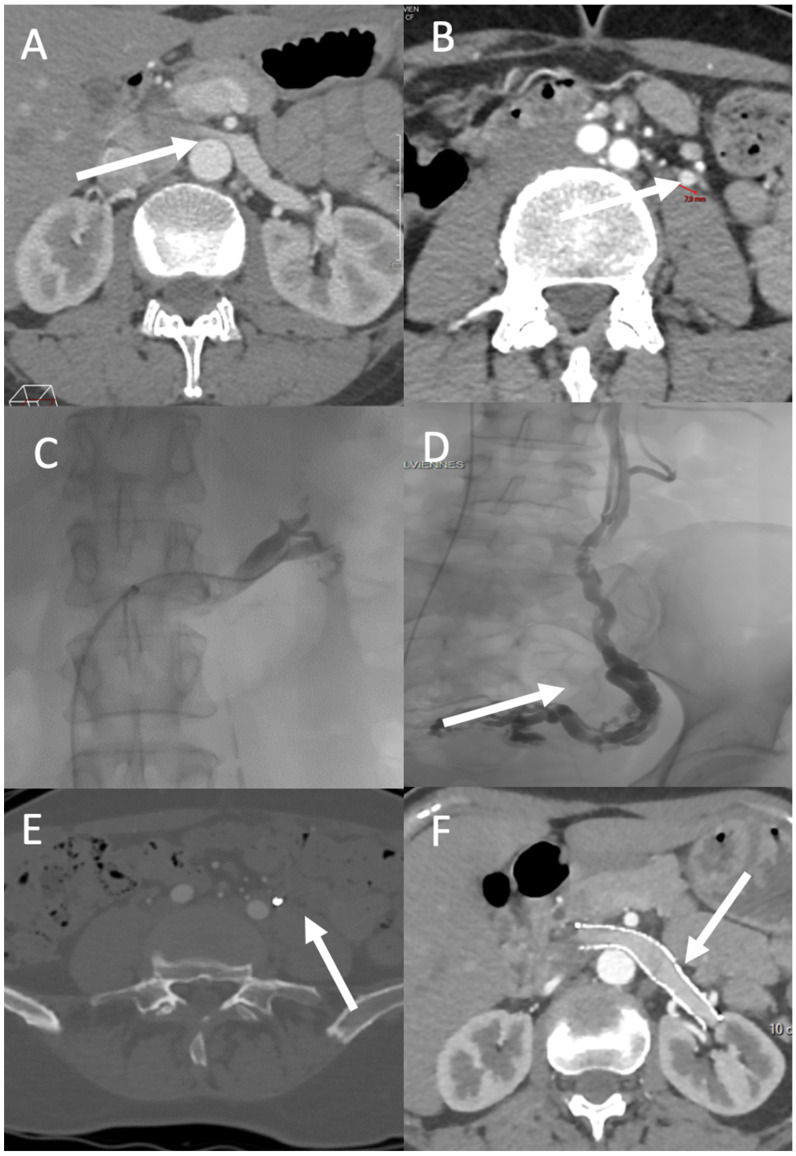
PVD affecting zones 1 and 2. A 49-year-old-woman with left flank pain (VAS pain score = 7), hematuria, and left chronic pelvic pain exacerbated by prolonged standing (VAS pain score = 5). (**A**) CT shows compression of the left renal vein over the abdominal aorta (arrow) and (**B**) a dilated left ovarian vein (diameter = 8 mm; arrow). Venography demonstrates attenuation over the abdominal aorta with a venous pressure gradient of 6 between the vena cava and left renal vein (**C**), and an incompetent left ovarian vein feeding a pelvic varicose vein (arrow) (**D**). Three months after treatment, CT shows successful embolization (using Onyx) of the left ovarian vein ((**E**), arrow) and patency of the left ovarian vein stent ((**F**), arrow). The left flank and chronic pelvic VAS pain scores decreased to 3 and 1, respectively.

**Figure 4 diagnostics-12-02337-f004:**
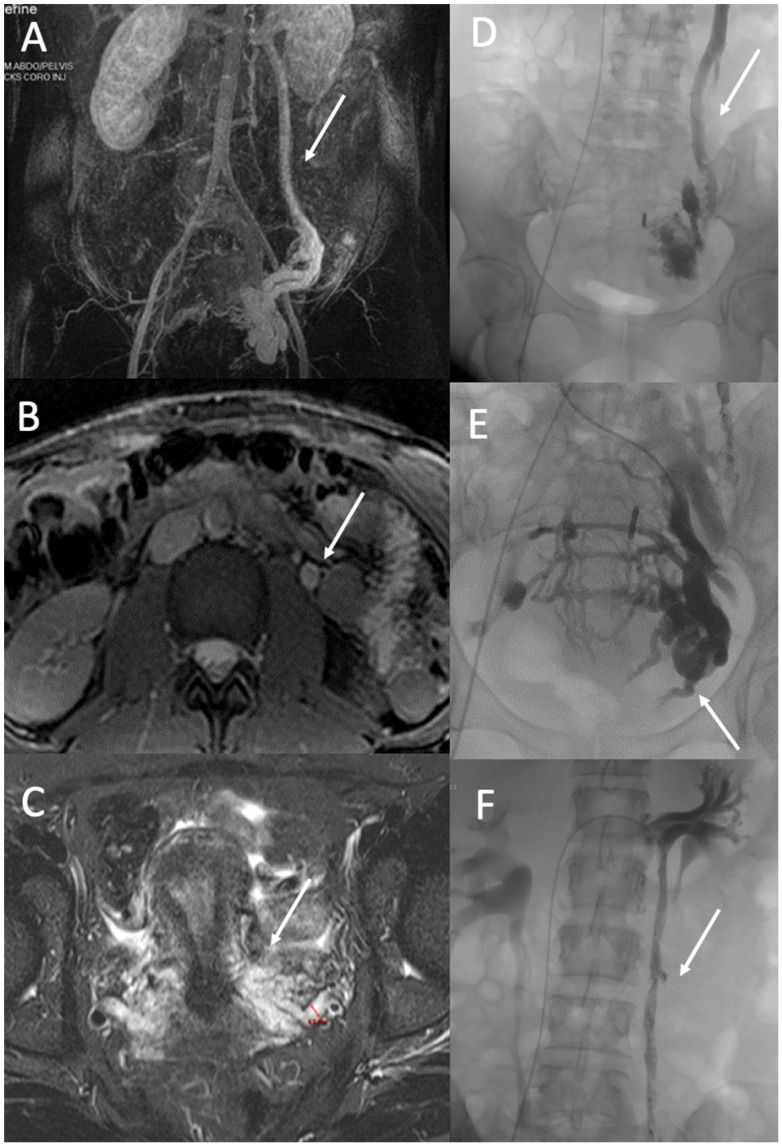
PVD of Zone 2. A 37-year-old woman with a typical pelvic congestion syndrome. (**A**). MR angiography image in the coronal plane shows incompetent and dilated left ovarian vein (arrow) and left pelvic veins with right periuterine varicose vein (**B**). True fast imaging with steady-state-free precession (TRUFI) demonstrated a dilated left ovarian vein (arrow) (10 mm) (**C**), and T2 STIR MR images (**C**) in an axial plane demonstrated dilated pelvic veins up to 8 mm (arrow). Phlebography shows incompetent left ovarian vein (arrow) (**D**) and incompetent internal iliac veins with periuterine and uterine varicose (arrow) (**E**). Treatment consisted of embolization with Onyx and Aetotoxisclerol of ovarian vein (arrow) (**F**) and pelvic varicosities.

**Figure 5 diagnostics-12-02337-f005:**
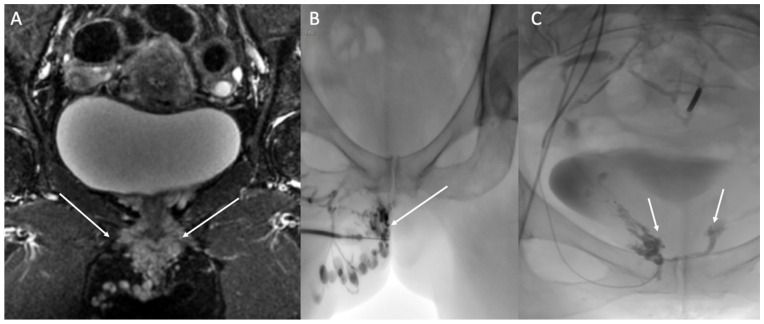
Painful right vulvar and perineal varices of pelvic origin with a pudendal leak point. (**A**) Coronal STIR MRI shows dilated pudendal veins (arrows). (**B**) Sclerotherapy of vulvar varicosities was performed through direct puncture. (0.5% polidocanol foam; arrow). (**C**) Occlusion of the right and left pudendal leak points (arrows).

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
