# Peer review of "Diagnosis and Management of Pelvic Venous Disorders in Females"

_diagnostics, 2022, doi:10.3390/diagnostics12102337_

Round 1

Reviewer 1 Report

1.       General comment. Although at the end of this submission it is stated that the manuscript has been proofread by a native speaker, some parts of the text are written in poor English or even are not understandable. The Authors should improve it.

2.       Abstract. In the abstract it is stated a study is presented. Actually, it is a review. Perhaps the Authors should use the modified Conclusion section as the abstract and discard Conclusions.

3.       In rare cases, gonadal and internal iliac venous insufficiency occur due to valvular incompetence or compression of the left renal or left common iliac vein. These PVDs are mainly caused by pregnancy…. This fragment is poorly understandable. Either these conditions are rare or are caused by pregnancy. Should be r3ewritten and clarified.

4.       Line 56. Regarding veins the term “incompetence” should be used instead of “incontinence” (the latter is used to describe the function of bladder and anus sphincters)

5.       Figure 2. Rectal point should be erased. Anatomically, in this area the portal and systemic venous systems connect. It has nothing to do with the main topic of the manuscript

6.       Figure 2. Proper anatomical terms should be used (see for example: https://books.google.pl/books?id=qsNbEAAAQBAJ&pg=PA13&lpg=PA13&dq=.+https://doi.org/10.1007/978-981-16-6206-5_2&source=bl&ots=GU1ds5OQgd&sig=ACfU3U132aiD-lUWII4V0i8ZP5hBkLdDzg&hl=pl&sa=X&ved=2ahUKEwiusKP5s-H4AhVProsKHQWBDYkQ6AF6BAgCEAM#v=onepage&q=.%20https%3A%2F%2Fdoi.org%2F10.1007%2F978-981-16-6206-5_2&f=false )

7.       Line 77. Should be: left renal vein

8.       Line 85. Use the term “ovarian vein” instead of “gonadal vein” since the discussion regards women

9.       Lines 98-111. In this study VAS and questionnaires described by the Authors were used, but is doesn’t mean that these questionnaires can be effectively used to diagnose pelvic congestion syndrome. This fragment should be revised

10.   Lines 112-119. This fragment is messy and difficult to understand, should be revised

11.   Lines 122-125. Actually leg  edema associated with the left iliac vein stenosis/occlusion results from pathology of this particular vein, and not from reflux coming from the renal or ovarian territory. Should be revised

12.   Lines 130-139. Proper anatomical and clinical terms should be used instead of hospital slang

13.   Lines 140-142. “The ratio between the diameter of the LRV at the renal hilum and AM [Please define on first use and complete sentence].  ???????????????

14.   Chapter 3.3. Proper anatomical terms should be used. Also, more details should be given considering how important are these anatomical features of atypical lower extremity veins. (see for example: https://books.google.pl/books?id=qsNbEAAAQBAJ&pg=PA13&lpg=PA13&dq=.+https://doi.org/10.1007/978-981-16-6206-5_2&source=bl&ots=GU1ds5OQgd&sig=ACfU3U132aiD-lUWII4V0i8ZP5hBkLdDzg&hl=pl&sa=X&ved=2ahUKEwiusKP5s-H4AhVProsKHQWBDYkQ6AF6BAgCEAM#v=onepage&q=.%20https%3A%2F%2Fdoi.org%2F10.1007%2F978-981-16-6206-5_2&f=false )

15.   Line 199. “through Giacominini’s anastomosis” What it is???

16.   Line 283. “evel of evidence = 2B0” ????

17.   Line 284. Proper pharmaceutical term should be used (eg. Polidocanol)

18.   Conclusions – these are not needed since it is a review paper

19.   References should be given in the format requested by the journal (see: Manuscript Preparation)

Author Response

  1. General comment. Although at the end of this submission it is stated that the manuscript has been proofread by a native speaker, some parts of the text are written in poor English or even are not understandable. The Authors should improve it.
  2. Abstract. In the abstract it is stated a study is presented. Actually, it is a review. Perhaps the Authors should use the modified Conclusion section as the abstract and discard Conclusions.

Authors: We modified it accordingly: This review aimed to explore the clinical and imaging modalities for diagnosing pelvic congestion syndrome

  1. “In rare cases, gonadal and internal iliac venous insufficiency occur due to valvular incompetence or compression of the left renal or left common iliac vein. These PVDs are mainly caused by pregnancy….”  This fragment is poorly understandable. Either these conditions are rare or are caused by pregnancy. Should be rewritten and clarified.

Authors: We modified it accordingly.

PVDs are mainly caused by pregnancy, and in rare cases, gonadal and internal iliac venous insufficiency occur due to valvular incompetence or compression of the left renal or left common iliac vein.

  1. Line 56. Regarding veins the term “incompetence” should be used instead of “incontinence” (the latter is used to describe the function of bladder and anus sphincters)

Authors: We modified it accordingly:

  1. Figure 2. Rectal point should be erased. Anatomically, in this area the portal and systemic venous systems connect. It has nothing to do with the main topic of the manuscript

Authors: This figure is from :Joël Constans. Les explorations vasculaires (SFMV) Société Française de Médecine Vasculaire, (CEMV) Collège des enseignants de médecine vasc, (CFPV) Collège Français de Pathologie Vasculaire. Elsevier Masson; 2014.

  1. Figure 2. Proper anatomical terms should be used (see for example: https://books.google.pl/books?id=qsNbEAAAQBAJ&pg=PA13&lpg=PA13&dq=.+https://doi.org/10.1007/978-981-16-6206-5_2&source=bl&ots=GU1ds5OQgd&sig=ACfU3U132aiD-lUWII4V0i8ZP5hBkLdDzg&hl=pl&sa=X&ved=2ahUKEwiusKP5s-H4AhVProsKHQWBDYkQ6AF6BAgCEAM#v=onepage&q=.%20https%3A%2F%2Fdoi.org%2F10.1007%2F978-981-16-6206-5_2&f=false )

Authors: Figure 2 come from a book.

Joël Constans. Les explorations vasculaires (SFMV) Société Française de Médecine Vasculaire, (CEMV) Collège des enseignants de médecine vasc, (CFPV) Collège Français de Pathologie Vasculaire. Elsevier Masson; 2014.

  1. Line 77. Should be: left renal vein

Authors: We modified it accordingly.

  1. Line 85. Use the term “ovarian vein” instead of “gonadal vein” since the discussion regards women

Authors: We modified it accordingly: we switched gonadal vein for ovarian vein in all the text.

  1. Lines 98-111. In this study VAS and questionnaires described by the Authors were used, but is doesn’t mean that these questionnaires can be effectively used to diagnose pelvic congestion syndrome. This fragment should be revised

Authors: these parameters help for the evaluation of PCS

  1. Lines 112-119. This fragment is messy and difficult to understand, should be revised

Authors: We modified it accordingly:

A recent study (15) suggested 10 clinical parameters for the evaluation of PCS using a Visual Analogue Scale (VAS) for pelvic pain, rated from 0 to 10 , dyspareunia, post-coital pain, periods, pain in the lower limbs, difficulty walking, aesthetic problems, impacts on work, psychology, and daily activities. Typically, PCS affects females aged 20–45, who are often multiparous

  1. Lines 122-125. Actually leg  edema associated with the left iliac vein stenosis/occlusion results from pathology of this particular vein, and not from reflux coming from the renal or ovarian territory. Should be revised

Authors: We deleted leg edema.

  1. Lines 130-139. Proper anatomical and clinical terms should be used instead of hospital slang

Authors: We modified it accordingly           Extrinsic compression of the common iliac vein by the right common artery and the spine called May and Turner Syndrome contribute to a secondary PSC, and leading to edema and lower limb venous claudication.

  1. Lines 140-142. “The ratio between the diameter of the LRV at the renal hilum and AM [Please define on first use and complete sentence].”  ???????????????

Authors: We modified it accordingly:Ratios ranging from 4.0:1 to 5.0: 1 are considered significant for Nutcracker syndrome

  1. Chapter 3.3. Proper anatomical terms should be used. Also, more details should be given considering how important are these anatomical features of atypical lower extremity veins. (see for example: https://books.google.pl/books?id=qsNbEAAAQBAJ&pg=PA13&lpg=PA13&dq=.+https://doi.org/10.1007/978-981-16-6206-5_2&source=bl&ots=GU1ds5OQgd&sig=ACfU3U132aiD-lUWII4V0i8ZP5hBkLdDzg&hl=pl&sa=X&ved=2ahUKEwiusKP5s-H4AhVProsKHQWBDYkQ6AF6BAgCEAM#v=onepage&q=.%20https%3A%2F%2Fdoi.org%2F10.1007%2F978-981-16-6206-5_2&f=false )

Authors: Sorry it was a mistake. Pudendal and not pudental.

  1. Line 199. “through Giacominini’s anastomosis” What it is???

Authors: the Giacomini vein (GV) is defined as a branch of cranial extension of the small saphenous vein (SSV) that connects the SSV with the posterior thigh circumflex vein (PTCV)

  1. Line 283. “evel of evidence = 2B0” ????

Authors: Sorry it was a mistake: 2B, not 2B0.

  1. Line 284. Proper pharmaceutical term should be used (eg. Polidocanol)

Authors: We modified it accordingly. polidocanol (Aetoxisclerol, Kreussler, Germany)

  1. Conclusions – these are not needed since it is a review paper

Authors: We deleted the conclusion.

  1. References should be given in the format requested by the journal (see: Manuscript Preparation)

Authors: We modified it accordingly.

Reviewer 2 Report

This is a well written review but the authors should also mention the role of laparoscopy in identifying the causes of PVD. 

Several studies have shown that laparoscopy is able to accurately identify the causes of CPP in patients, including PCS. 

More over some studies determined to be superior.

Author Response

Authors: We added: Laparoscopic has a sensibility of 40% in the detection of pathologies associate with PCS (28), however pelvic MRI is superior (14).

  1. Juhan V. Chronic pelvic pain: An imaging approach. Diagn Interv Imaging. 2015;96:997–1007.
  2. Arnaoutoglou C, Variawa RS, Zarogoulidis P, Ioannidis A, Machairiotis N. Advances of Laparoscopy for the Diagnosis of Pelvic Congestion Syndrome. Medicina (Kaunas). 2021;30;57:1041. doi: 10.3390/medicina57101041

Round 2

Reviewer 1 Report

Figure 2 is misleading and incorrect from anatomical point of view. Important connections between pelvic and lower extremity veins along the veins of round ligament and the inferior gluteal veis are poorly depicted in this picture. Rectal venous plexuses do not directly connect to the pelvic veins surrounding female reproductive organs. Besides, no such a wide vein as has been depicted in the picture can be found in humans.

If the picture indeed has been copied from another book, it should be properly citied and the permission from the copyright owner should be obtained

Correct anatomy of this region can be found at:

https://books.google.pl/books?hl=en&lr=&id=qsNbEAAAQBAJ&oi=fnd&pg=PA13&ots=GU1fxdRPi6&sig=BSy23pug6tnTQBPh_D1He1wkakU&redir_esc=y#v=onepage&q&f=false

Author Response

Dear: We delete this figure 2. 

regards

Round 3

Reviewer 1 Report

1. The term “nutcracker syndrome” should be written with small letters (not Nutcracker syndrome). This should be corrected throughout the text
2. There should be “May-Thurner syndrome”, not “May and Thurner syndrome”. This should be corrected throughout the text
3. Page 2, Line 64. Instead of „causing vulvar…” should be “results in vulvar…”
4. Page 3, line 100. Should be peri- , not per-
5. Page 3 line 102. Instead of „are common” should be “are similar”
6. Page 3 line 109. Instead of „greater” should be “great”
7. Page 4, line 122. Instead of „Is there significant” should be “Is there a significant”
8. Page 4, lines 133-135. The sentence is incomplete and with strange (editorial?) commentary. Needs correction
9. Page 4, line 138. Instead of „pressure readings” should be “pressure measurements”
10. Page 4, line 151. Instead of „in a study” should be “in the study”
11. Page 4, line 157. Instead of „a sensitivity” should be “the sensitivity”
12. Page 5, lines 172. Instead of „laparoscopic” should be “laparoscopy”
13. Page 5, lines 172. Instead of „associate” should be “associated”
14. Page 5, lines 173. Instead of „superior” should be “of a better diagnostic value”
15. Page 5, lines 177. Instead of „such be” should be “should be”
16. Page 5, lines 181. Instead of „inferior gluteal vein” should be “superior gluteal veins”
17. Page 5, lines 184. The abbreviation STIR should be explained
18. Page 5, lines 194. The term Giacomini’s anastomosis should be explained (anatomically, there is the Giacomini vein, still it is the vein localized intrafascially and joining the great and the small saphenous veins, thus is doesn’t seem to be this ‘anastomosis’). Needs clarification
19. Page 5, lines 200-202. This sentence deals with the arteries. Is it actually correct????
20. Page 5, line 210. Instead of „and or” should be “and/or”
21. Page 5, line 219. Instead of „considered the best” should be “considered to be the best”
22. Page 6, line 224. “55” – days? weeks? years?
23. Page 8, line 234. Instead of „pressure venous gradient” should be “venous pressure gradient”
24. Page 8. Lines 240-250. Company names, city, country of all devices should be provided.
25. Page 8, line 248. “Injections are performed…” Injections of what?
26. Page 10, line 282. Instead of „on iliac” should be “in iliac”
27. Page 10, line 284. Instead of „has proven” should be “has been proven”
28. Page 10, line 294. Instead of „has also proven” should be “has also been proven”
29. Page 10, line 312. Instead of „sclerosis can be addressed” should be “sclerotherapy can be done”
30. Page 10, line 316. Instead of „echography” should be “ultrasonographic control”
31. Page 10, line 317. Instead of „sclerosis is addressed by” should be “sclerotherapy is performed with”
32. Page 11, line 320. B) should be: Sclerotherapy of vulvar varicosities was performed through direct puncture.
33. Page 11, line 321. Instead of „sclerosis” should be “occlusion”
34. Page 11. Authors Contributions. This should be completed
35. Page 11. Funding. This should be completed
36. Page 11. Institutional Review Board Statement. This should be completed
37. Page 11. Informed Consent Statement. This should be completed
38. Page 11. Conflict of Interest. This should be completed

Author Response

  1. The term “nutcracker syndrome” should be written with small letters (not Nutcracker syndrome). This should be corrected throughout the text

Authors: We write it accordingly.

  1. There should be “May-Thurner syndrome”, not “May and Thurner syndrome”. This should be corrected throughout the text

Authors: We write it accordingly.

  1. Page 2, Line 64. Instead of „causing vulvar…” should be “results in vulvar…”

corrected throughout the text

Authors: We write it accordingly.

  1. Page 3, line 100. Should be peri- , not per-

Authors: We write it accordingly.

  1. Page 3 line 102. Instead of „are common” should be “are similar”

Authors: We write it accordingly.

  1. Page 3 line 109. Instead of „greater” should be “great”

Authors: We write it accordingly.

  1. Page 4, line 122. Instead of „Is there significant” should be “Is there a significant”

Authors: We write it accordingly.

  1. Page 4, lines 133-135. The sentence is incomplete and with strange (editorial?) commentary. Needs correction

Authors: We don’t understand what is strange.

  1. Page 4, line 138. Instead of „pressure readings” should be “pressure measurements”

Authors: We write it accordingly.

  1. Page 4, line 151. Instead of „in a study” should be “in the study”

Authors: We write it accordingly.

  1. Page 4, line 157. Instead of „a sensitivity” should be “the sensitivity”

Authors: We write it accordingly.

  1. Page 5, lines 172. Instead of „laparoscopic” should be “laparoscopy”

Authors: We write it accordingly.

  1. Page 5, lines 172. Instead of „associate” should be “associated”

Authors: We write it accordingly.

  1. Page 5, lines 173. Instead of „superior” should be “of a better diagnostic value”

Authors: We write it accordingly.

  1. Page 5, lines 177. Instead of „such be” should be “should be”

Authors: We write it accordingly.

  1. Page 5, lines 181. Instead of „inferior gluteal vein” should be “superior gluteal veins”

Authors: We write it accordingly.

  1. Page 5, lines 184. The abbreviation STIR should be explained

Authors: STIR is already explained P4, Line 172: STIR sequences (short TI inversion recovery)

  1. Page 5, lines 194. The term Giacomini’s anastomosis should be explained (anatomically, there is the Giacomini vein, still it is the vein localized intrafascially and joining the great and the small saphenous veins, thus is doesn’t seem to be this ‘anastomosis’). Needs clarification

Authors: we modified it: and through Giacominini’s veins, intersaphenous anastomosis.

We add this reference : 32.          Georgiev M, Myers KA, Belcaro G. The thigh extension of the lesser saphenous vein: from Giacomini’s observations to ultrasound scan imaging. J Vasc Surg. 2003 Mar;37(3):558–63t.

  1. Page 5, lines 200-202. This sentence deals with the arteries. Is it actually correct????

Authors: it was a mistake pudendal veins.

  1. Page 5, line 210. Instead of „and or” should be “and/or”

Authors: We write it accordingly.

  1. Page 5, line 219. Instead of „considered the best” should be “considered to be the best”

Authors: We write it accordingly.

  1. Page 6, line 224. “55” – days? weeks? years?

Authors:  We modified it: up of 45 months.

  1. Page 8, line 234. Instead of „pressure venous gradient” should be “venous pressure gradient”

Authors: We write it accordingly.

  1. Page 8. Lines 240-250. Company names, city, country of all devices should be provided.

Authors: We add it accordingly

  1. Page 8, line 248. “Injections are performed…” Injections of what?

      Authors:  Phlebography are performed during Valsalva maneuvers.

  1. Page 10, line 282. Instead of „on iliac” should be “in iliac”

Authors: We write it accordingly.

  1. Page 10, line 284. Instead of „has proven” should be “has been proven”

Authors: We write it accordingly.

  1. Page 10, line 294. Instead of „has also proven” should be “has also been proven”

Authors: We write it accordingly.

  1. Page 10, line 312. Instead of „sclerosis can be addressed” should be “sclerotherapy can be done”

Authors: We write it accordingly.

  1. Page 10, line 316. Instead of „echography” should be “ultrasonographic control”

Authors: We modify it accordingly: ultrasound guidance.

  1. Page 10, line 317. Instead of „sclerosis is addressed by” should be “sclerotherapy is performed with”

Authors: We write it accordingly.

  1. Page 11, line 320. B) should be: Sclerotherapy of vulvar varicosities was performed through direct puncture.

Authors: We write it accordingly.

  1. Page 11, line 321. Instead of „sclerosis” should be “occlusion”

Authors: We write it accordingly.

  1. Page 11. Authors Contributions. This should be completed

Authors: We write it accordingly.

  1. Page 11. Funding. This should be completed

Authors: We have no funding.

  1. Page 11. Institutional Review Board Statement. This should be completed

Authors: We write it accordingly.

  1. Page 11. Informed Consent Statement. This should be completed

Authors: We write it accordingly.

  1. Page 11. Conflict of Interest. This should be completed

Authors: We write it accordingly.